# The Participation of Microglia in Neurogenesis: A Review

**DOI:** 10.3390/brainsci11050658

**Published:** 2021-05-18

**Authors:** Diego R. Pérez-Rodríguez, Idoia Blanco-Luquin, Maite Mendioroz

**Affiliations:** 1Neurophysiology Department, Complejo Hospitalario de Navarra-IdiSNA (Navarra Institute for Health Research), 31008 Pamplona, Navarra, Spain; drpr25@gmail.com; 2Neuroepigenetics Laboratory-Navarrabiomed, Complejo Hospitalario de Navarra-IdiSNA (Navarra Institute for Health Research), Universidad Pública de Navarra (UPNA), 31008 Pamplona, Navarra, Spain; idoia.blanco.luquin@gmail.com; 3Department of Neurology, Complejo Hospitalario de Navarra-IdiSNA (Navarra Institute for Health Research), 31008 Pamplona, Navarra, Spain

**Keywords:** microglia, inflammation, adult neurogenesis, age, neurodegenerative diseases

## Abstract

Adult neurogenesis was one of the most important discoveries of the last century, helping us to better understand brain function. Researchers recently discovered that microglia play an important role in this process. However, various questions remain concerning where, at what stage, and what types of microglia participate. In this review, we demonstrate that certain pools of microglia are determinant cells in different phases of the generation of new neurons. This sheds light on how cells cooperate in order to fine tune brain organization. It also provides us with a better understanding of distinct neuronal pathologies.

## 1. Introduction

Neurogenesis is the formation of new neurons from neural stem cells and is a major event in the development of the neural system. Although neurogenesis is a process that fundamentally takes place in the pre- and postnatal periods, the synthesis of new neurons in mammals is maintained during adult life in certain brain niches [1]. Adult neurogenesis is crucial to consolidate memory and learning and may be altered as a result of aging, neurodegenerative diseases, and other pathological conditions, such as status epilepticus (SE) [2]. Interestingly, neurogenesis is strongly influenced by the action of brain-resident immune cells. In particular, microglia have been revealed as a central actor in the production, maturation, and integration of new neurons into existing circuitry. In this review, we summarize the current knowledge about the relevance of the microglia in adult neurogenesis.

For this purpose, we describe what microglia are and their actions in the human brain. We then discuss the process of neurogenesis in healthy individuals and in pathological conditions, highlighting the importance of microglia in both situations. Finally, we present our conclusions regarding these topics. A better knowledge of the biological processes in which microglia are involved and their effects on neurological diseases will help to improve the clinical management of these diseases.

## 2. Adult Neurogenesis

The term “neurogenesis” refers to the production of new neurons. This process occurs mainly in the embryogenesis period, during the generation of the cells of the organism. It was thought that neurogenesis only occurred in this stage of life, but it has been demonstrated that it occurs at other ages in both animals and humans [3,4]. We currently consider there to be two types of neurogenesis: embryonic and early postnatal neurogenesis and adult neurogenesis. The first leads organisms to create a global structure, whereas adult neurogenesis has a more restricted role: it is essential for modulating the functions of the hippocampus, a structure implicated in learning, memory, and behavior [5]. In addition, we can find adult newborn neurons in the olfactory bulb, which is related to odorant discrimination and olfactory learning [6] and discrimination learning and long-term memory [7]. Adult neurogenesis is also required for stress resilience [8].

Environmental (physical exercise, stress, inadequate diet, alcohol, etc.) and intrinsic signals are of great importance in this process, as is discussed hereafter. These factors can produce modifications that play a major role in maintaining NPCs and dictating their lineage commitment through the spatial and temporal expression of key regulators. Moreover, the release and uptake of extracellular signaling molecules, such as growth factors, neurotrophins, cytokines, and hormones, are under strict epigenetic control [9].

The finding that inflammation is closely related to human adult neurogenesis is an example of how determinant the environment is. For example, in vitro, microglia can produce IL-6, which causes apoptosis of neuroblasts; moreover, in the adult animal, inflammation as a result of bacterial endotoxin lipopolysaccharide (LPS) injection was found to reduce the production of new neurons in the neurogenic zone of the adult hippocampus [10].

Some years ago, it was discovered that neurogenesis occurs in adult humans. Investigators found this process in two localized areas: the SVZ, near the lateral ventricles, and the subgranular zone (SGZ), in the DG. These areas produce new neurons for the olfactory bulb and hippocampus, respectively [11].

This is possible because of the faculty of these cells to migrate. In the SVZ, they migrate across the RMS. The RMS was discovered in experiments with rats, and researchers, using IHC assays, recently found that it may also exist in humans [12].

However, this remains controversial, essentially because of the work published by Sorrells et al. in Nature in 2018: human adult neurogenesis was shown at undetectable levels in adults. There have also been other investigators who could not demonstrate the presence of human adult neurogenesis [13,14]. Nevertheless, this could be explained by technical limitations or the post-mortem interval of the selected subjects in the experiments [15].

## 3. Microglia

### 3.1. Generalities

In 1856, Rudolf Virchow described these cells for the first time, and in 1919, Pío del Río Hortega (a disciple of Santiago Ramón y Cajal) distinguished between astrocytes, microglia, and oligodendrocytes in brain non-neural cell populations [16]. Microglia are essentially the resident mononuclear phagocytes in the central nervous system (CNS) [17]. However, there are other mononuclear phagocytes in the CNS, such as macrophages in the perivascular space, but these originate separately to microglia: they are blood cells derived from progenitors in the bone marrow.

Human brain microglia constitute about 10% of all the cells in the CNS. They are regularly distributed throughout the CNS and originate from primitive macrophages in the yolk sac, which migrate to the neural tube [17]. In humans, microglial cells can be detected at 13 weeks of gestation and are a self-maintaining population that persists for months. Despite the belief that all tissue macrophages derive from monocytes, using parabiosis and fate mapping approaches, Hashimoto et al. proved that tissue macrophages such as microglia are mainly produced from primitive progenitors as a result of macrophage-colony stimulating factor (M-CSF) and granulocyte macrophage (GM)-CSF action [18]. CSF-1 signaling is also important for these cells. This factor is secreted by neurons, and it binds to its receptor in microglia (CSF1R). It has been reported in several studies that genetic defects in this receptor drastically reduce the number of microglial cells [17,19]. Moreover, in experiments with mice, a lack of this receptor results in a lower density of microglial cells. Furthermore, IL-34 is important for this, because this cytokine can also bind CSF1R, producing the same effect [20].

### 3.2. Functions

Microglia have different functions depending on the stage of life, CNS region, and environment (health and disease). However, ultimately, they act by sensing and regulating the environment of the CNS, eliminating certain structures such as pathogens, dead cells, protein aggregates, and others. They secrete cytokines and neurotrophic factors for that purpose and for other immune responses. Moreover, during life, they contribute to neurogenesis, neuronal circuit shaping, vascular formation, and the remodeling and maintenance of homeostasis [21].

A substantial part of microglial function depends on the expression of specific receptors, such as immune receptors and receptors for neurotransmitters. We can consider that pattern-recognition receptors (PRRs) recognize pathogen-associated molecular patterns (PAMPs) and tissue damage-associated molecular patterns (DAMPs). The latter stimulate microglia for the phagocytosis of apoptotic cells, protein aggregates, and lipoprotein particles [22]. For example, they can express a number of receptors, such as CD36 and SR1 [23,24], and the tyrosine kinase receptors Tyro3, Axl, and Mer. The aforementioned three receptors allow microglia to opsonize apoptotic cells, exposing phosphatidylserine (PS) [25]. They also express receptors for neurotransmitters as microglia are engaged in intense communication with neurons and glia. For instance, microglia carry receptors for NMDA (N-methyl-D-aspartic) and AMPA (alpha-amino-3-hydroxy-5-methyl-4-isoxazolepropionic), which in turn inhibit TNF (tumor necrosis factor)-alpha, limiting the inflammatory environment. Nevertheless, other neurotransmitters, such as substantia P, facilitate the transition to an inflammatory microglial state [26]. Finally, microglia are the only cells in the CNS that express CX3CR1 (C-X3- chemokine receptor type 1), a receptor for fractalkine and interleukin-8 (IL-8), which acts as a potent chemotactic factor for neutrophils [27].

Moreover, they can act as antigen-presenting cells (APCs), as they have the lysosomal machinery required and can stimulate memory T lymphocytes [28].

By collaborating with astrocytes, they are able to prune weak synapses. Astrocytes produce TGF-beta, which causes neurons to express C1q. Then, C1q marks the synapses for microglia pruning [29]. They also collaborate with astrocytes for the elimination of prion proteins; to this end, astrocytes release MFGE8 (milk fat globule-EGF factor 8 protein), which tags PrPSc (scrapie prion protein) [30].

### 3.3. Classification

In 1919, Pío del Río Hortega proposed the existence of microglial subtypes. Classically, we consider there to be two types of microglia: activated or resting microglia, based on observations of fixed microscope slides (resting microglia had a ramified shape, and activated microglia were the least ramified) [31]. Activated microglia have an ameboid morphology, and their processes phagocytize apoptotic cells, weak synapses, and proteins. Resting microglia have a ramified morphology with a lot of processes and branches. These processes involve other cells of the CNS, such as neurons or astrocytes. They enter an activated state in response to injuries, inflammation, and epileptic activity, for example, and acquire the aforementioned ameboid shape. The activated state is related with phagocytosis and an inflammatory environment [22].

In addition, there are other classifications related to activated microglia: M1 and M2. M1 are microglia in an inflammatory state, also known as a neurotoxic state, and they are capable of producing cytokines such as Il-6 or TNF-alpha; M2 are microglia in an anti-inflammatory state, producing Il-4 and Il-10 cytokines, among others [32] (Figure 1, Table 1 and Table 2).

Epigenetics is also fundamental in distinguishing between several types of microglia cells. For example, miR-124 is involved in the microglia resting state, and it is downregulated if microglia are exposed to inflammatory cytokines [33].

Differences between males and females regarding microglial cells have also been reported. For example, males have a higher density of microglia in the preoptic area, as experiments with mice show. Furthermore, male mice have more microglia in the cortex, hippocampus, and amygdala at an early postnatal age. This implies that in the event of a neonatal infection, male mice may produce more inflammatory cytokines, which can cause long-term deficits in cognition and memory in adulthood [34].

These are the classical microglia classifications, but this system is not sufficient to explain microglia in vivo [35]; for example, in models of neurodegeneration, we can find both M1 and M2 states at the same time. Moreover, many different subtypes of microglia have been described to date, with apparently different functions [36]. Interestingly, there is a lot of diversity in the expression of functional receptors (neurotensin, serotonin, vasopressin, etc.) and in the morphology of these cells. Thus, it seems that the different subpopulations of microglia are much more complex than initially thought. Accordingly, there are different properties and functions for particular types of microglia, and they can adopt distinct phenotypes in response to diverse stimuli and insults [36].

In summary, beyond the classical classification of microglia, there are different pools of microglia, which might each display intrinsic properties. For Tan et al., regional heterogeneity of microglia is a possible result of their environment, particularly the interaction with neurons or neural progenitor cells; however, intrinsic mechanisms are also possible, such as different origins and migration in different waves into the brain [37].

## 4. Role of Microglia in Neurogenesis

Adult neurogenesis and developmental neurogenesis are characterized by different features. Activated microglia regulate normal embryonic neurogenesis by phagocytosis and by secreting molecules such as nitric oxide (NO) and inflammatory cytokines. However, these molecules may be deleterious for adult neurogenesis. In vitro, microglia can produce IL-6, which leads to apoptosis of neuroblasts [38]. However, they can also produce IL-10, which has the opposite effect [39]. Most experiments indicate that microglial cells are neuroprotective. Nevertheless, a relatively small disturbance in CNS homeostasis can contribute to the progression of neurodegeneration, and microglia play an important role in this. For example, in most neurodegenerative diseases, microglia are stimulated with immune factors and Toll-like receptors (TLRs), and this has an impact on the capacity of microglia to phagocytize neurons and protein complexes [40].

Increasing evidence suggests that microglia are determinant cells of neurogenic niches, such as the SVZ and SGZ of the DG in the hippocampus. They have the capacity to guide the differentiation of precursor cells to neurons. An in vitro study showed that neural progenitor cells cultured in a microglia-conditioned media gave rise to a higher proportion of neurons because of soluble factors released by microglia [41].

There is heterogeneity in microglia in neurogenic areas. However, in murine models, different researchers have reported that microglia in these zones are significantly less involved than microglia from adjacent areas [42]. Significantly increased CD68 expression in microglia was also reported in the olfactory bulb of adult wild-type rats [43]. Furthermore, a microglial population expressing Clec7a was shown to be restricted to the DG by immunohistochemistry (IHC) [44].

Several relevant experiments involve microglia cells and neurogenesis. Of these, the work of Kreisel et al. in 2019 is of note [44]. Using IHC, quantitative real-time PCR, and RNA sequencing methods on neural cells extracted from mice brains, they concluded that “microglia cells play an essential role in survival of newly formed neuroblasts, for basal hippocampal neurogenesis and for vascular endothelial growth factor (VEGF)–induced neurogenic enhancement”.

These authors showed that microglia in neurogenic niches have characteristics and properties that differ from those of microglial cells located in other parts of the nervous system, and these allow them to fulfill functions related to neurogenesis. For example, they used a transgenic system for conditional VEGF induction through which they achieved an improvement in hippocampal neurogenesis. They concluded that DG microglia were the only susceptible microglia subpopulation involved in widespread local proliferation and activation.

### 4.1. Stages of Adult Neurogenesis and the Role of Microglia 

There are two paths that adult neural stem cells (ANSCs) may take: they can produce new astroglia under inflammatory conditions or become TAPCs (transient amplifying progenitor cells), neuroblasts, and new neurons, through proliferation, surveillance, differentiation, migration, and maturation [45] (Figure 2). Relevant molecules and regulatory factors, including Pax6, Tbr2, Sox2, and Prox1, participate in all these processes [46,47,48,49]. ANSCs can differentiate into neurons, astrocytes, and oligodendrocytes in cell cultures [50]. In this article, we refer exclusively to neural production.

#### 4.1.1. Proliferation

Proliferation is amplification by the division of the ANSCs. These are the type 1 cells, they are multipotent and can be in a quiescent or an active state depending on the environment and molecular signaling. For instance, Notch1 stimulates the quiescent state, preventing excessive activation [51]. BMP (bone morphogenetic protein) signaling is also important in this phase. BMPs are necessary for the quiescent state of ANSCs, but they are also important for differentiation and maturation of granule neurons [52]. Microglia are able to activate latent ANSCs through CX3CR1 in the hippocampus of mice engaged in exercise [53].

#### 4.1.2. Surveillance

Newly formed TAPCs constitute type 2 cells; however, not all of them survive. Many die as a result of apoptosis and are subsequently phagocytized by activated microglia. Indeed, when an aberrant proliferation occurs, activated microglia cells can phagocytize the poorly formed cells, as we describe above [54].

Apoptosis of newborn cells occurs in the first 1 to 4 days of life. This apoptosis may be directly induced by inflammatory cytokines released by activated microglia cells, which then phagocytize them [54]. However, it has been reported that, when mice are treated with annexin V, which binds to the PS receptor, presumably blocking phagocytosis, the number of apoptotic cells increases and neurogenesis in the SGZ is reduced by decreasing the survival of neuroblasts [55].

#### 4.1.3. Differentiation

Unlike ANSCs, type 2 cells or TAPCs have limited self-renewal properties [45]. Despite this, they express Ascl1, which promotes their proliferation and differentiation into neurons [56]. At first, they do not have the characteristic radial morphology, i.e., they are TAPCs in an earlier phase (type 2a). When they acquire a neural phenotype, they constitute type 2b cells, and they form migratory neuroblasts [57].

#### 4.1.4. Migration

Migratory neuroblasts are type 3 cells. They move from the neurogenic niche, e.g., the SGZ, to other zones in the hippocampus through intermediate progenitors (IPs), which support and guide later migrating NSCs by Notch signaling and filopodia-like leading processes. These processes allow them to contact mitotic progenitors, including other IPs and NSCs, which facilitates their migration [58].

Microglia also have a role in this step. The experiments of Ribeiro et al. showed that SVZ microglia depletion has a diminishing effect on neuroblast migration 4–7 days later [42].

#### 4.1.5. Maturation

Migratory neuroblasts differentiate into immature neurons, which are characterized by a transient expression of the calcium binding protein calretinin (CR) [59]. Adult-born neurons express calbindin (CB) once they are mature. This also correlates with different morphological characteristics, as is observed in rodents and in human-adult neurogenesis [59].

By detecting doublecortin (DCX)+ cells using IHC studies, we generally find CR+ cells at the hilar border of the granule cell layer (GCL) or the SGZ. These cells have smaller soma and an elongated morphology, with two primary neurites oriented parallel to the SGZ. CB+ cells, however, are placed in deeper positions in the GCL and have larger soma and an oval morphology, with one primary apical neurite oriented perpendicular to the SGZ [60].

Adult newborn cells have to compete with mature granule cells, which constitute pre-existing neural circuits. They have to invade and replace pre-existing synapses [61]. It is hypothesized that this is possible because adult-born immature neurons are more excitable than mature cells, so they are more efficient in generating action potentials, even with weak glutamatergic inputs. This probably makes them more capable of gaining control of the synaptic sites [62]. In addition, in support of this hypothesis, recent studies have shown that microglia preferentially phagocytize weak or less active synapses [63]. Microglia also eliminate redundant neurons that do not establish functional circuits and aberrant newborn progenitor cells in SE (this state promotes neurogenesis) in animal models [2].

However, it should be noted that activity-dependent synaptic pruning by microglia was found during postnatal development. Various recent studies support the role of microglia in remodeling synaptic connections in the adult hippocampus [64,65]. Indeed, there is direct evidence to suggest that microglia play a role in regulating synaptic competition between newborn neurons and mature granules cells in the adult mice; these adult-born neurons that have replaced pre-existing synapses participate in certain types of memory and learning [66].

Nguyen and his group further explored this topic. They discovered that microglia remodel the extracellular matrix (ECM) driven by the cytokine Il-33, which allows them to promote synapse plasticity and the consolidation of long-term memories. Several ECM proteases (Adamts4, Mmp14, Mmp25, and Ctsc) are associated with this process. Microglial contact with newborn neurons as a result of Il-33 has been correlated with the formation of new dendritic spines and the induction of spine head filopodia [65].

### 4.2. Recent Discoveries

Díaz-Aparicio et al. (2020) [67] found that the microglial phagocytosis of neural progenitors is essential for neurogenesis. They realized that chronic impairment of microglial phagocytosis decreases adult hippocampal neurogenesis, whereas an acute microglial phagocytosis impairment transiently increases it.

This group reported that phagocytic microglia actively maintains neurogenesis by sensing apoptosis. Interestingly, they proposed that microglial phagocytosis provides a negative feedback loop that is necessary for the long-term maintenance of adult hippocampal neurogenesis.

In addition, they found that the secretome (containing metabolites, miRNAs vesicles, extracellular vesicles, etc.) of phagocytic microglia reduced the most mature neuroblast subpopulation at 28 days in vivo. Therefore, it seems that microglia could regulate neurogenesis through the phagocytosis secretome. However, the secretome of inflammatory microglia is not enough to stimulate microglia to phagocytize neural stem cells [67].

There are some similarities between inflammatory and phagocytic secretomes, such as the content of IL-1, IL-6, and TNF cytokines. Nevertheless, experiments with the secretome of inflammatory microglia did not trigger a reduction in the survival of NPCs [67].

Microglia play a key role in modulating adult neurogenesis, which is not limited to the clearance and removal of cells. They are also involved in neuroprotection and neurodegeneration, and their presence in neurogenic niches, displaying specific characteristics and properties, has been proven. They guide the differentiation of precursor cells into neurons throughout the whole process; they guide the microglial phagocytosis of neural progenitors, which is as essential for neurogenesis; and they remodel synaptic connections in the adult hippocampus, helping to establish functional circuits, which promotes consolidation of long-term memories. Nevertheless, further studies are needed to fully determine microglia involvement in neurogenesis.

### 4.3. Role of Certain Types of Microglia in Neurogenesis

Our understanding of microglial heterogeneity has improved in recent years, and several types were discovered using imaging techniques, genetic approaches, and transcriptomic single-cell/bulk RNA analyses [68].

#### 4.3.1. CD11c-Microglia

The proportion of this type of microglia varies during development. It is one-fifth of the total in the postnatal brain but represents less than 3% in juvenile and adult mice. Gene ontology (GO) enrichment analysis revealed that CD11c-microglia express genes associated with neurogenic and myelinogenic processes in the neonatal brain [69]. In fact, CD11c microglia is present at high concentrations in regions of the brain related to neurogenesis, such as the SVZ [70].

During neuroinflammation, CD11c microglia are the major cellular source of insulin-like growth factor-1 (Igf1), secreted phosphoprotein 1 (Spp1), glutathione peroxidase 3 (Gpx3), and galectin 1 (Lgals1), which are critical for neurodevelopment, myelination, and neurogenesis. Galectin-1, which is encoded by Lgals1, stimulates neural regeneration and suppresses microglial activation [69].

This type of microglia expresses the majority of myelinogenic Igf1 in the developing brain. IGF1 is necessary for neurodevelopment both in humans and in mice [70].

CD11c+ microglia may also contribute to axonal organization in the CNS through expression of axonal guidance signals such as Cxcl12, Ntn1, Plxna2, and Sema7a [70]. They may stimulate neuronal and neurite outgrowth and astrocytogenesis through expression of Ptn [71].

#### 4.3.2. TREM2-Microglia

It is known that *TREM2* is only expressed in the brain by microglia. The number of *TREM2*-expressing microglia cells is highest in the cingulate cortex and lateral entorhinal cortex and much lower in the hypothalamus and habenula [72]. This type of microglia is present in low concentrations in the SVZ and in high concentrations in the cerebral cortex and hippocampus [37].

*TREM2* is a receptor that takes part in the activation of microglial phagocytosis and is involved in neuronal cell survival and neurogenesis [73]. TREM2 microglia is related to Igf1 and other genes with a significant participation in neurogenesis. It plays a role in the maintenance of microglial levels of STAT6, which functions as a key transcription factor for IL-4 (an anti-inflammatory cytokine). Indeed, low TREM2 levels are associated with neurodegeneration [74].

## 5. What Happens in Certain Conditions

Neurogenesis decreases in many pathological conditions (such as in an inflammatory environment) because of microglia, which act as a regulator of this process. In those conditions, it could result in aberrant neurogenesis.

### 5.1. Inflammatory Environment (Injuries, Vascular Damage, Epilepsy, SE, Neurodegenerative Processes)

Neuroinflammation is a cellular and molecular response to brain pathologies or injuries. It induces the activation of resident microglia and local invasion by immune cells. An inflammatory environment induces a substantial decrease in the production of new neurons, as has been reported in various experiments with mice [75]. Activated microglia acquire an ameboid shape, which facilitates the phagocytosis of neural progenitors, and release inflammatory cytokines, such as Il-6, TNF-alpha, and Il-1beta. Finally, these amplified microglial responses exacerbate neurodegenerative processes [38,75]. A number of inflammatory molecules are involved in these activities.

TNF-alpha, for example, was shown to play a detrimental role in neural differentiation and to decrease neurogenesis when added to ANSC cultures. The suppression of neurogenesis by TNF-alpha may constitute a mechanism to preserve brain integrity when aberrant neurogenesis occurs, as Pérez-Domínguez et al. proposed. This occurs through the activation of TLR9, a receptor through which microglia are able to recognize the self-DNA of degenerating neurons as DAMPs, and this reduces aberrant neurogenesis through the production of TNF-alpha [76].

Cytokine Il-1beta reduces neurogenesis at any time point during the lifespan. It induces cell cycle arrest, mediated by the tumor suppressor p53, which causes the inhibition of neural precursor cell proliferation. This has been proven in experiments in vitro and in vivo, and its effect can be prevented by deleting the type I Il-1 receptor (IL-1R1) [77]. Moreover, Il-6 can reduce neurogenesis, decreasing the proliferation, survival, and differentiation rate of neural precursor cells. Il-6 also promotes ANSC differentiation and astrogliogenesis by the regulatory activity of microRNA miR-155 [78].

In addition, it was recently reported that another inflammatory cytokine (IL-33) may also play an important role in microglia-mediated newborn neuron integration in an experience-dependent manner (synapse remodeling is essential to encode experiences, such as fear memories, into neuronal circuits, and Il-33 participates in this) [65].

Some years ago, it was well accepted that microglia phagocytize dead or dying cells but not living cells. However, microglia can also engulf living cells during development, inflammation, and in neuropathological conditions. This is called primary phagocytosis, in contrast to secondary phagocytosis, which refers to engulfing dying and dead cells [79].

An inflammatory environment can be created by a pathological insult, such as infection or brain injury. In this regard, thanks to such work as that developed by Dennis et al. in 2013, we know that microglial activation occurs as a result of certain brain insults, such as hepatic encephalopathy [80].

Epileptic seizures can also induce microglial activation in the hippocampus [76]. In animal models of temporal lobe epilepsy, SE stimulates adult neurogenesis at first. However, these cells have abnormal differentiation, ectopic positions, and aberrant dendrites and axons. This creates an inflammatory environment that activates microglia, which selectively engulf extensively proliferating cells after SE to suppress the number of newborn cells in order to maintain homeostasis [2].

In addition, we can find an inflammatory environment due to stroke, Alzheimer’s disease or brain injury, which is related to neurogenesis diminution through microglial activation.

Neuroinflammation can also be produced by stress, by an inadequate diet, or by alcohol consumption. This latter, for example, may cause the production of pro-inflammatory factors, such as TNF-alfa and reactive oxygen species (ROS) [81].

#### Ways to Decrease Neuroinflammation

Therefore, an inflammatory environment reduces adult neurogenesis. Nevertheless, this inflammation may be decreased in several ways, thereby promoting neurogenesis. For example, physical exercise has a global anti-inflammatory effect on the organism, due to an increase in Il-10 and glucocorticoids [82].

In several experiments with mice, we can see that acute wheel running promotes ANSCs expansion, probably by recruiting quiescent cells and stimulating their division. Type 2 cells are also highly responsive to wheel running [83]. Moreover, this activity also increases the amount of insulin-like growth factor 1 (IGF-1), inducing a neuroprotective microglia subtype [84]. Wu et al. (2007) and Lee et al. (2014) obtained evidence regarding the benefits of aerobic exercise in neurogenesis. Wu et al. observed that mice that perform treadmill exercise are more capable of counteracting the suppressive effects of LPS in neurogenesis [85]. Lee et al. also showed that physical exercise reduces proinflammatory conditions, improving neurogenesis [86]. Furthermore, Gebara et al. monitored microglia and the proliferation of adult hippocampal progenitor cells in association with voluntary running and aging (physiological conditions known to increase and decrease adult neurogenesis, respectively). They showed that the number of NPCs in the DG was strongly affected by microglia [87].

### 5.2. Aging

Neurogenesis decreases with age, and, in humans, it occurs more rapidly than in other mammals. Dennis et al., from the University of Sydney and the University of New South Wales in Kensington, Australia, state that “from middle age neurogenesis occurs at relatively low and perhaps negligible levels”. The number of neural progenitors decays with age in the SVZ and SGZ, and by four years of age, the density of these cells in the neurogenic niches is not higher than in the adjacent zones [9].

Other studies using IHC and immunofluorescence (IF) in combination with unbiased stereology also showed that age-related neurogenesis decay is faster in humans than in other mammals [88].

Curtis et al. demonstrated the presence of an RMS in humans by means of IHC studies [89]. Sanai et al. used Ki67 and DCX markers to show that SVZ neurogenesis and migration through RMS decays at 2 years of age and is almost non-existent in adulthood [90]. However, Knoth et al., using DCX and PCNA markers, evidenced that neurogenesis occurs in the SGZ in adults up to 100 years old [91]. There are still conflicting results on the timing of this diminution and exactly how it occurs in the human brain [13].

In any case, even decreased adult neurogenesis is essential for the aging brain and any further reduction is related with neurodegenerative diseases [13]. In connection with this, it has been reported that increasing neuronal Il-33 release in individual neurons improves certain structural and functional correlates of aging in the brain [61].

Moreover, microglia develop an enhanced immune response during aging, probably because of increased peripheral inflammation or a loss of the integrity of the blood–brain barrier, which may also be associated with telomere shortening [92]. In addition, the inflammatory response related to activated microglia is linked to an alteration in lysosomal biogenesis due to senescence. This lysosomal dysfunction occurs spontaneously with ageing and causes the exacerbation of inflammatory responses as compared to young subjects [13].

So-called “dark microglia” have recently been discovered in the hippocampus, cerebral cortex, amygdala, and hypothalamus of adult and aged mice. Dark microglia strongly express CD11b. They have a robust association with capillaries and the basal lamina, so it is thought that they may have an involvement in the blood–brain barrier. They display an electron-dense cytoplasm and nucleoplasm, which gives them a dark appearance in electron microscopy [93].

### 5.3. Neurodegenerative Diseases

In neurodegenerative diseases, neurogenesis is affected. These diseases reduce the self-renewal capacity of ANSCs and the number of adult newborn neurons. It has been postulated that the reduction in adult-born neuron development may contribute to cognitive decline [94].

However, it has also been demonstrated that, in these cases, organisms try to enhance neurogenesis in order to avoid neurodegeneration. For example, in the human brain tissues of patients who died with Huntington’s disease, a significant increase in the levels of proliferating cell nuclear antigen (PCNA), beta III-tubulin (a neuronal marker), and glial fibrillary acidic protein (GFAP- a glial cell marker) can be found [95]. Moreover, there is evidence for raised numbers of BrdU-labeled proliferating cells in Parkinson’s disease models [96]. Nevertheless, it seems that these attempts to maintain the neuronal population are not good enough, and this stimulation of adult neurogenesis is not capable of preventing neurodegeneration. In the following section, we describe in more detail what happens in AD.

#### Neurogenesis in AD

AD is the most common form of adult dementia, and according to the amyloid hypothesis [97], it is caused by the accumulation of Aβ protein plaques and Tau protein in the brain. It is considered that these deposits lead to a chronic inflammation of the CNS. Different meta-analyses have revealed that AD patients have elevated rates of TNF-alpha and Il-1beta, and, as we described above, these proinflammatory cytokines reduce adult neurogenesis through microglia cell activity.

*TREM2* microglia seem to be fundamental for eliminating Aβ plaques in AD. Rare genetic variants of *TREM2* increase the risk of developing late-onset AD. In this neurodegenerative condition, a proinflammatory environment is found around βA plaques, and microglia phagocytosis and immune responses have a relevant function [34]. In addition, in trials testing novel treatments for AD, such as humanized antibodies, microglia cells have been shown to play a fundamental role. They contribute to the therapeutic effects of these treatments by phagocytizing opsonized plaques. Thus, in *Trem2*-deficient mice, the efficacy of these antibodies was reduced [98].

Another subgroup of microglia, dark microglia, lack heterochromatin patterns and, in neurons, this is related to diseases such as AD or schizophrenia [99]. Thus, dark microglia are rarely expressed in healthy young mice but are strongly present in models of AD and in conditions such as chronic stress and CX3CR1 knockout. As an example, the number of these cells quadruples in young adult mice after 2 weeks of chronic unpredictable stress, as compared with controls [100].

AD eventually generates symptoms such as episodic memory loss, disorientation, and apraxia, and manifestations such as a reduction in hippocampal volume. As in other neurodegenerative conditions, there are also changes in the olfactory capacities. This is interesting because we can find adult neurogenesis in the SGZ (from which adult-born neurons are incorporated into the hippocampus) and in the SVZ (from which adult-born neurons are incorporated into the olfactory bulb through the RMS) [93].

Nevertheless, as mentioned in the previous section, in neurodegenerative diseases, there are processes that try to reduce this degeneration. Therefore, studies on patients and mouse models have shown increased levels of early neuronal differentiation markers, such as TUC-4, DCX, and polysialic acid-neural cell adhesion molecule (PSA-NCAM), in the hippocampi of AD brains [101,102]. Furthermore, increased Il-33, a molecule that is implicated in how microglia respond to inflammatory conditions and is related to the number of spine head filopodia; increased dendritic spine plasticity; and neuronal activity has been found to be associated with improved cognition in mouse AD models [103].

## 6. Conclusions

Neurogenesis can occur in adult humans in certain areas, such as the SVZ and SGZ. Microglia cells play a pivotal role in this process. These cells can promote normal neurogenesis or stop it. This is important in the SGZ for learning and the incorporation of memories and in situations where aberrant neurogenesis is produced, such as during SE. Neurogenesis is also altered in a number of brain pathologies, particularly in neurodegenerative diseases, mainly due to the generation of inflammatory environments. This involves the activation of microglia cells, which phagocytize ANSCs in neurogenic niches.

In summary, further understanding of these processes will allow neuroscientists to better manage the physiopathology of neurodegenerative diseases and, ultimately, to find new targets to prevent neurodegeneration. This will in turn help to reduce the pain that these pathologies cause to patients and their relatives.

## Figures and Tables

**Figure 1 brainsci-11-00658-f001:**
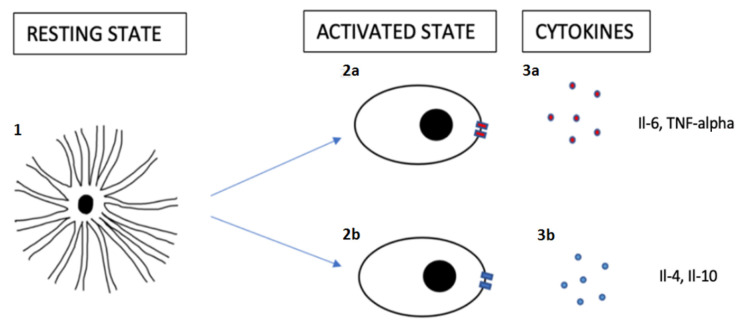
Microglial subtypes. **1**: Resting microglia with ramified morphology with a lot of processes and branches. miR-124 is involved in the resting state of microglia. **2**: Activated microglia with ameboid morphology. **2a**: M1, activated microglia associated with an inflammatory state. **2b**: M2, microglia in an anti-inflammatory state. **3a**: Cytokines that can produce an inflammatory environment, such as Il-6 or TNF-alpha. **3b**: Cytokines with anti-inflammatory properties (Il-4, Il-10, for example).

**Figure 2 brainsci-11-00658-f002:**
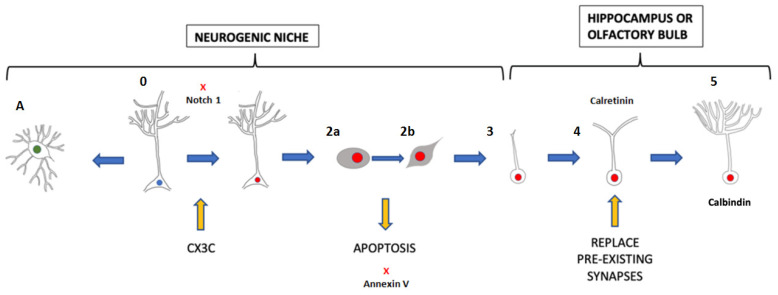
Process of neurogenesis. **0**: ANSC in a quiescent state. **1**: Type 1 cell. ANSC-activated. **2**: Type 2 cell (differentiating **2a** and **2b** because of their phenotype). TAPC. **3**: Type 3 cell. Migratory neuroblast. **4**: Immature neuron. **5**: Mature neuron. A. Astrocyte. The yellow arrows represent the participation of microglia in this process: microglia are able to activate latent ANSCs by the CX3C chemokine receptor (CX3CR1) in the hippocampus of mice engaged in exercise. They also participate in neurogenesis, inducing apoptosis in some type 2 cells, before the microglia phagocytize them. In addition, newborn cells have to compete with mature neurons and microglia collaborate at this point because they phagocytize weak or less active synapses. Notch1 stimulates the quiescent state, preventing the excessive activation of ANSCs. Annexin V is able to reduce apoptosis and may increase the number of apoptotic TAPCs. Migratory neuroblasts differentiate into immature neurons, which are characterized by a transient expression of calretinin.

**Table 1 brainsci-11-00658-t001:** Summary of cytokines, showing their most relevant functions.

Interleukins (IL)	Communication between Leukocytes (Especially LTh)
Lymphokines	generated by lymphocytes
Monokines	generated by monocytes
Interferons (IFNs)	antiviral responses
Chemokines	chemoattraction between cells
Colony Stimulating Factors	growth of cells

**Table 2 brainsci-11-00658-t002:** Summary of cytokines, showing which are pro-inflammatory and anti-inflammatory.

Cytokines	Examples
Pro-inflammatory cytokines	TNF-alpha, IL-1, IL-2, IL-6, and IL-8
Anti-inflammatory cytokines	IL-4, IL-10, IL-11, IL-13

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
