# Peer review of "The Participation of Microglia in Neurogenesis: A Review"

_brainsci, 2021, doi:10.3390/brainsci11050658_

Round 1

Reviewer 1 Report

This review article discusses the role of microglia in adult neurogenesis. It gives an overview of microglia, neurogenesis and then the role that these cells play in that process. There is also information on how certain conditions might result in aberrant neurogenesis via alterations in microglia. While this is an interesting topic, there is quite a bit of reorganization that needs to be done. In additional, there are several missing studies that should be added to provide evidence that microglia participate in the various stages of adult neurogenesis. I have outlined my more specific comments below.

Organization:

In general, many of the discussed topics go back and forth. It would be best to discuss a single topic clearly in one section and not in multiple sections. 

There is quite a bit of information about the role of microglia in the different stages of adult neurogenesis. Yet, it is followed up with an entire section on the role of microglia in neurogenesis. This should be reorganized so that it is clear.

In the section labeled “microglia supporting neurogenesis”, CD11b and TREM2 microglia are introduced, but then they aren’t discussed in sufficient detail until the section “role of microglia in neurogenesis”. I’m not sure what the difference between these two sections are. It may be best to combine.

The last section on “what happens in certain conditions” reads much more clearly than the rest of the paper. Yet, there is still some information that would be better included in the earlier parts of the paper (human neurogenesis, etc).

Other comments:

There is quite a bit of studies on the role of microglia in each stage of adult neurogenesis. How microglia participate should be much more clearly worded and cited. For example, microglia participate in adult neurogenesis via synapse remodeling. While Nguyen et al., 2020 work is mentioned, it doesn’t address the study at all which is about how microglia remodel the ECM and that’s “how” they allow for more synaptic integration. This is just an example of one study, but should be done throughout. 

Overall, there is quite a bit of reference to specific receptors, cytokines, etc, with little information on each one. This should be revised. There in general is a lot of information about this, but the actual “how” for how microglia participate in adult neurogenesis is not very clear.

The Spalding et al article is cited for the role of adult neurogenesis in hippocampal functions. Yet, this paper is looking at adult human neurogenesis. There is a lot of work showing that adult neurogenesis participates in behaviors associated with the hippocampus (e.g. learning and memory, pattern separation, anxiety regulation, social behavior, etc). Please choose a different citation.

Along the lines of adult neurogenesis in humans. There is a mention of it, but the papers showing it aren’t cited (or cited later in the context of aging). In recent years this has been made a controversy. These citations should be added and discussed.

Under the “neurogenesis” section, line 205, it states that “environmental and intrinsic signals are of great importance to this process” Yet, there is no mention of possible signals (running, stress, etc) in this section. Instead, how these signals alter epigenetics is discussed. I think the specific signals should be mentioned. There is also a variety of ways that these “signals” can alter neurogenesis via microglia, not just epigenetics. It seems strange to mention that one without the others. It should either be removed or elaborated on to discuss all of the possibilities.

In line 31, it uses the word “synthesis” of new neurons. I think this is misleading. Microglia participate at various stages in the production, maturation, and integration of new neurons into existing circuitry. This eludes to proliferation only.

Genetic deletion of CSF1R is mentioned to reduce microglial numbers. There is no mention in a growing body of work that uses pharmacologics to inhibit CSF1R and that is known to deplete microglia.

I think the section on microglia and the microbiota should be deleted. It seems out of place and doesn’t add anything to the paper.

In line 152, I would delete the RMS. Those cells are born in the SVZ and migrate along the RMS to populate the OB. They are not a different population of cells. If they are, please cite.

There is a typo on line 365. It reads “TREM2-microglia is It known that…”

In general, there are a lot of reviews cited.

In the section on aging, there is mention of the human literature. There is an extensive literature on aging in the rodent brain. Given that the majority of this review is focused on animal models, I think this should be discussed here and not just the human literature. Instead, the human adult neurogenesis literature (including the controversy) might be work discussing in the adult neurogenesis section.

Reviewer 2 Report

1. This manuscript is a well prepared text but still need a complete proofreading in English grammar and some spellings.
2. In the section of "Neurogenesis in AD", the author should emphasize the initial contribution of microglia to neuroinflammation. But some of the proinflammatory cytokines in late neuroinflammation did not come from microglia.
3. Line 499, Abeta protein plaque, not betaA protein.
4. The 'CD11c-microglia', 'TREM2-microglia', and 'Dark microglia' had been verified as overlap subgroups using single cell RNAseq analysis. The author should not neglect these improtant progresses in microglial differentiation and subgrouping.
5. Some statement are not appropriate scientific language, eg. line 404-405. The author should use plain scientific accent to illustrate the data. 
6. Section 5.2 'Age' is not common word, the author should use ageing or aging.
7. Line 497, the author listed a section 'Neurogenesis in AD' without Category ID.
8. The author should follow the format rule of the journal. The Figure 1 had no figure title. The Table 1 can be divided into 2 tables.
9. The Category ID of the section are disordered. The author may combine two manuscripts without careful editing work. eg. Line 149, '2.3.1. Microglia supporting neurogenesis' is not a continuous Category ID.
10. The major defect of the manuscript is that the author never reveal the role of microglia in neurogenesis, even at a hypothesis level. The figures, tables and references did not support the title of the review.

Round 2

Reviewer 1 Report

While the manuscript has improved quite a bit from the initial submission, there is still some comments that need to be addressed.

1). The authors states “adult neurogenesis has a more restricted role: it is essential for modulating the functions of the hippocampus.” This is true for hippocampal adult neurogenesis, but necessarily for the OB neurogenesis. Adult-born neurons in the hippocampus also participate in more functions behaviorally than just learning and memory. The roles of adult neurogenesis in each of these brain regions (OB and hippocampus) should be mentioned.

2). For clarity and organizational purposes, I think the neurogenesis section should be eliminated and just discuss adult neurogenesis. Embryonic and postnatal neurogenesis is really not relevant to the review and the role of microglia in that process is not discussed. I would remove paragraph starting at line 50 and then combine the other paragraphs in to the adult neurogenesis section. The other factors (e.g. environment enrichment, etc) is discussed as altering developmental neurogenesis. There is quite a body of literature showing that these factors alter adult neurogenesis and should be mentioned there instead.

3). I think the classical definitions of microglia (resting and activated) are a little misleading and should be mentioned. Even ramified microglia are never resting – that work should be cited.

4). In the “Stages of adult neurogenesis and the role of microglia” there is a lot of repetitiveness that is mentioned in the specific stages. I think it is confusing here and should be eliminated to just state the stages of adult neurogenesis and that microglia participate at each of the stages. The following sections can be used to describe the specific role of microglia in more detail.

5). What role do C11-c microglia participate in neurogenesis? All the section states is that they are expressed in high levels in neurogenic regions?

6). The “recent discoveries” 4.3 section under “certain types of microglia is confusing. What certain type of microglia is it discussing? Shouldn’t that section be incorporated under how microglia shape different aspects of adult neurogenesis (e.g. section stages of adult neurogenesis and role of microglia)?

7). Line 366-367. This sentence is not true for all pathological conditions (e.g., epilepsy, which is mentioned to increase adult neurogenesis) so it should say in neurogenesis decreases in some or many pathological conditions. Or even state, that it results in aberrant neurogenesis. 

8). Line 412 mentions that physical exercise has anti inflammatory effects. Then it the next paragraph it talks about the role of microglia in physical exercise increases in neurogenesis. I think the minocycline and ginseng sentence should be removed (Lines 413-414) unless there is more discussion of whether those pharmacologics alter adult neurogenesis via microglia. Without it, it doesn’t add information to the role of microglia in neurogenesis.

9). Along this thought, physical exercise is listed under inflammatory conditions. That is misleading as it is the opposite of such. Instead, I would remove it and discuss inflammatory conditions in more detail. Unless you want to have a section under the certain conditions that is more positive (e.g. physical exercise, environmental enrichment, etc). The only inflammatory really mentioned in this section is epilepsy. Yet, there is lots of studies from stroke, Alzheimer’s disease, brain injury etc that could be mentioned. Even briefly if they all point to the same thing (aka increased inflammation and reduced adult neurogenesis as a result). It also at the end mentions other states that produce inflammation (stress, etc), but doesn’t mention that those also often result in fewer new neurons. For a reader that is not familiar with the literature, this link would not be made.

10). I think that the authors should consider a title change to be grammatically correct. Maybe "The participation of microglia in neurogenesis: a review"

Author Response

Point 1: The authors states “adult neurogenesis has a more restricted role: it is essential for modulating the functions of the hippocampus.” This is true for hippocampal adult neurogenesis, but necessarily for the OB neurogenesis. Adult-born neurons in the hippocampus also participate in more functions behaviorally than just learning and memory. The roles of adult neurogenesis in each of these brain regions (OB and hippocampus) should be mentioned.

Response 1: We thank reviewer #1 for these instructive comments. We have included this relevant information (lines 49-52).

Point 2: For clarity and organizational purposes, I think the neurogenesis section should be eliminated and just discuss adult neurogenesis. Embryonic and postnatal neurogenesis is really not relevant to the review and the role of microglia in that process is not discussed. I would remove paragraph starting at line 50 and then combine the other paragraphs in to the adult neurogenesis section. The other factors (e.g. environment enrichment, etc) is discussed as altering developmental neurogenesis. There is quite a body of literature showing that these factors alter adult neurogenesis and should be mentioned there instead.

Response 2: The reviewer #1 is correct on this point. We have eliminated the neurogenesis section and we have combined the information on it with “Adult neurogenesis”.

Point 3:  I think the classical definitions of microglia (resting and activated) are a little misleading and should be mentioned. Even ramified microglia are never resting – that work should be cited.

Response 3: We thank reviewer #1 for this relevant suggestion. We have added this important information for the manuscript (lines 135-137).

Point 4:  In the “Stages of adult neurogenesis and the role of microglia” there is a lot of repetitiveness that is mentioned in the specific stages. I think it is confusing here and should be eliminated to just state the stages of adult neurogenesis and that microglia participate at each of the stages. The following sections can be used to describe the specific role of microglia in more detail.

Response 4: We thank reviewer #1 for this comment. We have modified the structure of this section, eliminating a paragraph with contents which were described on the following sections and we have included more information on the description of Figure 2 (lines 227-233).

Point 5: What role do CD11-c microglia participate in neurogenesis? All the section states is that they are expressed in high levels in neurogenic regions?

Response 5: Reviewer #1 is right at this point. Information about CD11-c microglia was incomplete in our manuscript, so we have written a new paragraph about this (lines 345-351).

Point 6: The “recent discoveries” 4.3 section under “certain types of microglia is confusing. What certain type of microglia is it discussing? Shouldn’t that section be incorporated under how microglia shape different aspects of adult neurogenesis (e.g. section stages of adult neurogenesis and role of microglia)?

Response 6: We thank reviewer #1 for this relevant suggestion. We have moved this section and now it is section 4.2 (line 303).

Point 7: Line 366-367. This sentence is not true for all pathological conditions (e.g., epilepsy, which is mentioned to increase adult neurogenesis) so it should say in neurogenesis decreases in some or many pathological conditions. Or even state, that it results in aberrant neurogenesis. 

Response 7: We thank reviewer #1 for this comment. We have modified this sentence (lines 365-367).

Point 8: Line 412 mentions that physical exercise has anti inflammatory effects. Then it the next paragraph it talks about the role of microglia in physical exercise increases in neurogenesis. I think the minocycline and ginseng sentence should be removed (Lines 413-414) unless there is more discussion of whether those pharmacologics alter adult neurogenesis via microglia. Without it, it doesn’t add information to the role of microglia in neurogenesis.

Response 8: The reviewer #1 is correct on this point. We have removed this sentence.

Point 9: Along this thought, physical exercise is listed under inflammatory conditions. That is misleading as it is the opposite of such. Instead, I would remove it and discuss inflammatory conditions in more detail. Unless you want to have a section under the certain conditions that is more positive (e.g. physical exercise, environmental enrichment, etc). The only inflammatory really mentioned in this section is epilepsy. Yet, there is lots of studies from stroke, Alzheimer’s disease, brain injury etc that could be mentioned. Even briefly if they all point to the same thing (aka increased inflammation and reduced adult neurogenesis as a result). It also at the end mentions other states that produce inflammation (stress, etc), but doesn’t mention that those also often result in fewer new neurons. For a reader that is not familiar with the literature, this link would not be made.

Response 9: We thank reviewer #1 for this relevant suggestion. We have created a new section (5.1.2.Ways to decrease neuroinflammation) and we have included lines 410-414 at the end of section 5.1.

Point 10: I think that the authors should consider a title change to be grammatically correct. Maybe "The participation of microglia in neurogenesis: a review"

Response 10: The reviewer #1 is correct on this point. We have changed the title.

hment

Reviewer 2 Report

The maunscript had been improved and could be published after English proofreading by native speaker.
